# Anti-EGFR Therapy Induces EGF Secretion by Cancer-Associated Fibroblasts to Confer Colorectal Cancer Chemoresistance

**DOI:** 10.3390/cancers12061393

**Published:** 2020-05-28

**Authors:** Colleen M. Garvey, Roy Lau, Alyssa Sanchez, Ren X. Sun, Emma J. Fong, Michael E. Doche, Oscar Chen, Anthony Jusuf, Heinz-Josef Lenz, Brent Larson, Shannon M. Mumenthaler

**Affiliations:** 1Lawrence J. Ellison Institute for Transformative Medicine, University of Southern California, Los Angeles, CA 90033, USA; colleenmgarvey@gmail.com (C.M.G.); roylau@usc.edu (R.L.); alyssams@usc.edu (A.S.); rensun@usc.edu (R.X.S.); efong@ellison.usc.edu (E.J.F.); mdoche@ellison.usc.edu (M.E.D.); oscarchen12@gmail.com (O.C.); ajusuf@usc.edu (A.J.); 2Division of Medical Oncology, Norris Comprehensive Cancer Center, Keck School of Medicine, University of Southern California, Los Angeles, CA 90033, USA; lenz@med.usc.edu; 3Department of Pathology and Laboratory Medicine, Cedars-Sinai Medical Center, Los Angeles, CA 90048, USA; Brent.Larson@cshs.org

**Keywords:** colorectal cancer, cancer-associated fibroblasts, epidermal growth factor, cetuximab, drug-resistance, tumor microenvironment

## Abstract

Targeted agents have improved the efficacy of chemotherapy for cancer patients, however, there remains a lack of understanding of how these therapies affect the unsuspecting bystanders of the stromal microenvironment. Cetuximab, a monoclonal antibody therapy targeting the epidermal growth factor receptor (EGFR), is given in combination with chemotherapy as the standard of care for a subset of metastatic colorectal cancer patients. The overall response to this treatment is underwhelming and, while genetic mutations that confer resistance have been identified, it is still not known why this drug is ineffective for some patients. We discovered that cancer-associated fibroblasts (CAFs), a major cellular subset of the tumor stroma, can provide a source of cancer cell resistance. Specifically, we observed that upon treatment with cetuximab, CAFs increased their secretion of EGF, which was sufficient to render neighboring cancer cells resistant to cetuximab treatment through sustained mitogen-activated protein kinases (MAPK) signaling. Furthermore, we show the cetuximab-induced EGF secretion to be specific to CAFs and not to cancer cells or normal fibroblasts. Altogether, this work emphasizes the importance of the tumor microenvironment and considering the potential unintended consequences of therapeutically targeting cancer-driving proteins on non-tumorigenic cell types.

## 1. Introduction

Colorectal cancer (CRC) has been well studied over the years, leading to a relative understanding of the genetics involved in disease progression and the identification and validation of promising drug targets. For example, supplementing chemotherapy regimens with cetuximab, a monoclonal antibody that targets epidermal growth factor receptor (EGFR), is now the standard of care for the *KRAS* wild-type subset of metastatic CRC patients. However, such treatment offers only modest benefits. Many genetic alterations have been identified as sufficient to confer resistance to cetuximab such as mutations in *KRAS*, *BRAF*, and alterations in the phosphoinositide 3-kinase/phosphatase and tensin homolog (PIK3CA/PTEN) pathway [1,2]. However, the mechanism of an estimated 10–30% of patients with initial clinical resistance remains unknown [2,3]. While this drug has been extensively investigated in terms of its effects on cancer cells, how this targeted agent affects the surrounding tumor microenvironment is still unknown.

In an era of precision medicine, biomarkers are used to optimize therapies and are thought to improve clinical endpoints. While the most studied and validated biomarkers are tumor cell intrinsic, the contributions of the surrounding microenvironment are increasingly well-recognized [4] and warrant further investigation. Cancer-associated fibroblasts (CAFs), the predominant cell type in the tumor stroma, have been implicated in various aspects of tumorigenesis including metastasis [5] and therapeutic resistance [6]. CAFs arise from multiple origins such as resident fibroblasts, epithelial cells, and distant bone marrow mesenchymal stem cells [7]. They lack mutations found within cancer cells [8] and are molecularly characterized by the expression of CAF-associated markers (such as αSMA, vimentin, fibronectin) [9]. They also express proteins fundamental to cellular processes, one of which is EGFR.

CAFs share a common environmental niche with cancer cells and collectively encounter EGFR inhibition during cetuximab treatment. Previous studies have identified contexts where the secretomes of stromal cells including CAFs [10,11] and B cells [12] are changed in response to chemotherapy and radiation treatments to confer resistance to the surrounding cancer cells. The effect cetuximab has on CAFs and its potential implications on cancer cell drug response have not previously been investigated. Here, we detail our findings that cetuximab treatment causes patient-derived CAFs isolated from three human CRC tumors to increase the secretion of EGF, which subsequently leads to increased resistance of CRC cancer cells to treatment.

## 2. Results

### 2.1. Cancer-Associated Fibroblasts (CAFs) Express Epidermal Growth Factor Receptor (EGFR) but Remain Viable during EGFR Inhibition

The influence of cancer therapeutics on non-tumorigenic cells is often overlooked. Given that CAFs are known to be involved in many aspects of tumorigenesis including drug resistance, we investigated whether these cells expressed EGFR, the target of cetuximab. Immunofluorescence staining of surgical tumor resections from CRC patients showed EGFR co-localized with αSMA, a marker used to identify CAFs, suggesting that CAFs do express EGFR (Figure 1A). Furthermore, CAFs isolated from these patient tumor tissues and cultured in vitro (confirmed to be CAFs through the expression of CAF-associated markers; Figure 1B, Appendix A), also expressed EGFR (Figure 1C). In contrast to CRC cells (DiFi and LIM1215), inhibiting EGFR signaling with cetuximab treatment does not alter overall CAF viability (Figure 1D).

### 2.2. CAFs Decrease Cancer Cell Sensitivity to Cetuximab

We previously established an imaging-based methodology that allows one to study the influence of drugs on heterogeneous cell populations while distinguishing between cell types [13,14]. Briefly, we quantified live and dead cell counts over time and then fit the data to an exponential growth model to determine net birth, death, and growth rates. This allows for readouts of cellular dynamics across time as well as distinguishing between cytotoxic (increased death rate) and cytostatic (decreased birth rate) effects of the drug. When we applied this approach to co-cultures of patient-derived CAFs and cancer cells at a starting ratio of approximately 1:1, the presence of CAFs prevented cancer cell death, even at high concentrations of cetuximab (Figure 2A,B; Appendix A). This result is comparable to in vitro cetuximab resistance due to *KRAS* mutations (Appendix A) [15]. Furthermore, when the starting fraction of CAFs-to-cancer cells was increased, reminiscent of CRC clinical stromal percentages (Appendix A), we observed a stronger protective effect against cetuximab, as evidenced by an increased growth rate of the cancer cells (Figure 2C). A minimum population of approximately 30% CAFs prevented cetuximab-induced death of cancer cells. CAF-driven increased growth in the untreated conditions was not dependent on CAF proportion (Appendix A).

Next, we sought to identify whether the CAF protective effect was dependent on physical cellular interactions or CAF-secreted factors. Keeping in mind that CAF secretomes may change in response to drug treatment, media were collected from untreated and treated CAFs. When added to cancer cell cultures, the conditioned media from cetuximab-treated CAFs (CMtx) provided more protection than untreated CAF conditioned media (CM) during cetuximab treatment (Figure 2D, Appendix A). This finding suggests that CAF secretomes change in response to cetuximab treatment, leading to the protection of cancer cells from the drug’s effects.

### 2.3. CAFs Secrete More Epidermal Growth Factor (EGF) in Response to Cetuximab Treatment

In order to identify secreted factors specific to cetuximab-treated CAFs, a cytokine array was performed to compare CM versus CMtx (Figure 3A,B; Appendix A). Surprisingly, the only common differentially expressed cytokine in the treated vs. untreated CAF samples was epidermal growth factor, EGF. This cetuximab-induced increase in CAF secretion of EGF was confirmed via enzyme-linked immunosorbent assay (ELISA), with at least a two-fold increase seen for each patient-derived CAF line (Figure 3C). This secretion pattern was not affected by culture media (Appendix A) and was sustained across five days, which was the longest time point tested (Appendix A). Moreover, this CAF effect occurred regardless of the cancer cell mutations found within the tumors that the CAFs were isolated from (Appendix A). We next wanted to verify whether this was a CAF-specific effect or a result seen across all cell types. CRC cell lines and patient-derived normal colon fibroblasts were found to secrete very low baseline levels of EGF and did not increase EGF secretion upon treatment with cetuximab (Figure 3C). Furthermore, EGFR inhibition by erlotinib, a small molecule inhibitor that binds to the intracellular tyrosine kinase domain, or treatment with oxaliplatin, a chemotherapy used to treat colorectal cancer, did not initiate increased secretion of EGF (Figure 3D). This suggests that increased EGF secretion by CAFs depends on cetuximab binding to the extracellular region of EGFR and not a general response to inhibition of the EGFR pathway or a general stress response.

### 2.4. Exogenous EGF Causes Cetuximab Resistance in 2D and 3D Cell Culture Models

Standard culture media for 2D immortalized cancer cell lines do not contain EGF. However, when supplemented with increasing concentrations of EGF, cancer cell growth rates increased in proportion with EGF concentration at each cetuximab dose (Figure 4A, Appendix A). We hypothesized that EGF-induced resistance to cetuximab results from sustained signaling through the MAPK pathway. In cancer cells, EGF stimulation increased levels of pERK1/2, whereas cetuximab treatment shut down this pathway, as evidenced by undetectable pERK1/2 levels. In contrast, co-treatment of cetuximab and EGF stimulation preserved MAPK signaling. Furthermore, pERK levels increased in correlation with increasing EGF concentrations (Figure 4B, Appendix A, Appendix A, Appendix A). This shows that pERK1/2 is rescued with the addition of EGF, even in the presence of cetuximab.

Patient-derived organoid models more accurately resemble patient tumors given their genetic and microenvironmental heterogeneity, although CAFs and other stromal cells are often not present. Culture media developed to support long term growth of 3D patient-derived organoids contain various supplements including EGF. Previous studies have warned about the potential bias these exogenous factors may impart in the context of drug response [16]. In order to translate our findings to a more physiologically relevant cancer model, we repeated our cetuximab and EGF experiments in *KRAS* wild-type patient-derived CRC organoids, ORG12620. When we lowered EGF concentration in the media (0.4 ng/mL from the previously defined 50 ng/mL), we restored cetuximab sensitivity in our CRC organoids (Figure 4C,D; Appendix A) with no significant decrease in overall viability in the untreated condition after five days (Appendix A). Furthermore, the addition of EGF during cetuximab treatment preserved MAPK pathway activity with pEGFR, pHER2, and pERK levels mirroring baseline levels (Figure 4E, Appendix A).

### 2.5. Secreted EGF from Cetuximab-Treated CAFs Is Sufficient to Render Cancer Cells Resistant to Cetuximab

To verify that EGF was the specific CMtx-factor that conferred resistance to cetuximab, we incubated CMtx with an EGF-neutralizing antibody (CMtx-EGF) (Appendix A), which led to cancer cell response to cetuximab through reduced cell viability. Specifically, cancer cells that were exposed to CMtx-EGF were re-sensitized to cetuximab at a level resembling baseline response (Figure 5A–C). The CMtx-induced resistance is likely to be due to sustained signaling through the MAPK pathway, as ERK is still active (Figure 5D, Appendix A). This supports the hypothesis that EGF in the CMtx media is causing resistance, as similar results were observed in cancer cells treated with exogenous EGF and cetuximab (Figure 4).

## 3. Discussion

Molecular targeting agents have significantly impacted the treatment of cancer; however, a large portion of the protein targets are expressed not only in cancer cells, but also in other cell types. There is limited research being done to investigate potential phenotypic responses to targeted agents, especially effects other than viability, which may occur in cells throughout the body including the stromal component of the tumor microenvironment. We discovered CAF secretion of EGF is increased in response to cetuximab and the presence of exogenous EGF results in cancer cell resistance to cetuximab treatment. While our studies focused on cetuximab treatment, analogous results are anticipated with panitumumab, an alternative monoclonal antibody targeting EGFR, which has similar clinical efficacy and toxicity profiles to cetuximab [17]. Future work will investigate the underlying mechanism leading to CAF secretion of EGF during extracellular inhibition of EGFR. The observation that EGF can outcompete an EGFR antibody in cell models has been previously reported [16,18,19,20], however, a source of EGF secretion from the stromal microenvironment in response to cetuximab treatment has not previously been identified.

The ratio of CAFs to cancer cells varies across patient tumors (Appendix A). If this ratio is low, it is likely that the concentration of EGF secreted by CAFs is not sufficient to rescue cancer cells from cetuximab treatment. However, as the tumor shrinks from cetuximab treatment, the ratio of CAFs will increase (since CAF viability is not affected—Figure 1C) and it is possible that EGF levels will be adequate for protection from cetuximab and therefore also be a cause of relapse to treatment. It has been shown that the stromal microenvironment changes over the course of cetuximab treatment. Of note, in patients with progressive disease, an increase in stromal abundance was observed when compared to baseline (i.e., prior to cetuximab treatment) [21]. This increase in stromal cells could be a culprit in treatment resistance, with the proportion of cancer cells to CAFs reaching a state where the secreted EGF levels are sufficient to sustain MAPK signaling in the presence of cetuximab. Combination treatment with MAPK inhibitors may be an attractive target to mitigate CAF-induced cetuximab resistance [22,23]. In recent years, there has also been a push to develop therapeutics that target CAFs [24]. We hypothesize that utilizing such drugs in combination with cetuximab may be another way to increase cetuximab efficacy. Furthermore, ongoing work is focusing on potential dual-targeting of receptor tyrosine kinases that may be activated in colorectal cancer cells in response to increased exogenous EGF.

There have been multiple clinical studies looking at biomarkers for cetuximab response that may supplement the current genetic alterations used for treatment stratification [25]. The CMS4 subtype of CRC tumors, which are characterized by a high stromal density [3], have been found to be prognostic for poor response to anti-EGFR treatment [26,27]. Furthermore, when looking at plasma levels of EGFR ligands, an increase in EGF levels from two weeks post-cetuximab treatment compared to initial treatment levels were significantly higher in non-responders [28]. Another independent study also identified a significant increase in EGF serum levels after cetuximab treatment, which corresponded to disease progression [29]. These clinical observations support our findings that EGF can confer cetuximab-resistance, with our results homing in on the stromal microenvironment as a significant culprit.

Most therapeutic agents used for treating cancer are given systemically and therefore have the potential to affect cells throughout the body. While this concept is considered extensively in the context of adverse side effects, the potential of one’s body contributing to better or worse overall response to the drug has just recently begun garnering attention. For example, microbiome composition is indicative of overall response to PD-1 based immunotherapy [30]. Our data suggest that CAF composition is important for cetuximab response, specifically highlighting EGF secretion by cetuximab treated CAFs as a previously unknown mechanism of resistance to anti-EGFR treatment in colorectal cancer.

## 4. Materials and Methods

### 4.1. Cell Culture and Reagents

DiFi and LIM1215 cancer cell lines were obtained from Dr. Alberto Bardelli (University of Torino) and cultured in Dulbecco’s Modified Eagle Media (DMEM) (Corning Inc., Corning, NY, USA) and Roswell Park Memorial Institute (RPMI) (Corning Inc., Corning, NY, USA), respectively, supplemented with 10% fetal bovine serum (FBS) (Gemini Bio, Sacramento, CA, USA) and 1% penicillin/streptomycin (P/S) (Gemini Bio, Sacramento, CA, USA) under standard laboratory conditions (5% CO_2_, 37 °C).

### 4.2. Primary Cell Culture: Human Tumor Organoids and Cancer-Associated Fibroblasts

Tumor tissues were received from colorectal cancer patients under Institutional Review Board (IRB) approval at the Norris Comprehensive Cancer Center of the University of Southern California (USC). All subjects gave their informed consent for inclusion before they participated in the study. The study was conducted in accordance with the Declaration of Helsinki, and the protocol was approved by the IRB Ethics Committee of USC (Protocol HS-06-00678; approval date 08-02-2019). Known tumor mutations and treatment data are detailed in Appendix A.

Patient-derived metastatic colorectal tumor organoids were developed following previously described methods [31]. Briefly, tumor tissue was digested with 1.5 mg/mL collagenase (MilliporeSigma, Burlington, MA, USA) and 20 μg/mL hyaluronidase (MP Biomedicals, Irvine, CA, USA) for 30 min at 37 °C, then separated through a 100 μm strainer (Corning Inc., Corning, NY, USA). Isolated cells were cultured in 3D using basement membrane extract gels to allow for tumor organoid formation. CAFs were separated from the same digested tumor tissue by culturing a fraction of cells on plastic tissue culture plates and letting the fibroblasts grow out over 1–2 passages. Cells were then verified as CAFs via qPCR and immunofluorescence staining for common CAF markers: α-smooth muscle actin (αSMA) (Abcam, Cambridge, MA, USA), vimentin (VIM) (Abcam, Cambridge, MA, USA), fibronectin (FN1) (Life Technologies, Grand Island, NY, USA), and fibroblast specific protein (FSP) (MilliporeSigma, Burlington, MA, USA) (Figure 1B, Appendix A, respectively). For all experiments, primary CAFs were used between passages 2 and 8.

### 4.3. Imaging Growth Rate Assays

Cells were seeded in four 384-well plates 24-h prior to treating with cetuximab (USC pharmacy, Los Angeles, CA, USA). On day 0, cells were treated with the drug at the desired concentration. Before imaging, cells were stained with 5 μg/mL Hoechst 33342 (nuclear dye) (Life Technologies, Grand Island, NY, USA) and 5 μg/mL propidium iodide (PI) (Life Technologies, Grand Island, NY, USA) to identify cells as live or dead, respectively. Individual plates were imaged on days 0, 2, 3, and 5 using the Operetta High Content Screening (HCS) system (PerkinElmer, Waltham, MA, USA)). Cells were then segmented based upon the nuclear dye using Harmony software (PerkinElmer, Waltham, MA, USA). In order to differentiate cell types in co-culture assays, morphological features were calculated and used to train a machine-learning algorithm to classify cells as either ‘CAF’ or ‘tumor,’ as described in Garvey et al. [13,14]. Propidium iodide intensity levels were calculated and cells were classified as ‘dead’ if their intensity was above the established threshold. Growth rates for each cell type were calculated as previously described [13,32] by fitting the live cell counts over time to an exponential growth model.

### 4.4. Collection and Processing of Conditioned Media

When CAF cultures reached approximately 80% confluent, media was changed to DMEM supplemented with 1% P/S and 10% FBS. Cells were treated with 1 μg/mL cetuximab or IgG isotype control and incubated for 72 h (unless otherwise specified). Media were collected, spun to remove debris, and stored at −80 °C. Media were thawed and incubated overnight with protein A/G agarose to remove remaining drug (CMtx) or IgG isotype control (CM). After separation the media from the agarose pellets, 0.5 μg/mL EGF neutralizing antibody (CMtx-EGF) or IgG isotope control (CMtx) was added and media were incubated at 37 °C for one hour.

### 4.5. Secretome Analysis

When cells were at approximately 70% confluence, culture media were replaced with FBS and P/S free DMEM for three days. This medium was then collected, spun down to remove debris, aliquoted, and stored at −80 °C. Frozen conditioned media were thawed and subjected to cytokine arrays (R&D systems, ARY022B) or ELISAs (R&D systems, DEG00), both following the manufacturer’s instructions. Clustering, overlap analysis, and visualization: Clustering analysis and visualization were performed in the R statistical environment (v3.6.0) [33] using the cluster package (v2.0.9) [34] and BPG (BoutrosLab.plotting.general) package (v5.9.2) [35]. Modified Z-scores were generated by gene-wise scaling of the cytokine array data by median and standard deviation. These values were subsequently clustered in heatmaps. Visualization of overlaps between groups was facilitated via the VennDiagram package (v1.6.20) [36].

### 4.6. Western Blotting

Cells were serum-starved overnight and treated with EGF or cetuximab for the time specified. Cells that were treated with EGF and cetuximab were incubated with cetuximab for 1 h prior to the addition of EGF for the specified time. Cells were harvested on ice and needle treated using RIPA buffer supplemented with protease and phosphatase inhibitors. The protein lysates (30 μg) were then resolved on 4–12% Bis-Tris gradient pre-cast gels (Invitrogen), transferred onto polyvinylidene difluoride (PVDF) membranes via semi-dry transfer. Immunoblotting was then performed with corresponding antibodies. Quantification of protein bands with densitometry was performed using FIJI ImageJ. The analysis was added to the uncropped western blot images included in the Appendix A (Appendix A).

### 4.7. Statistical Analysis

Unpaired t-tests were performed using GraphPad Prism version 8.0.0 (GraphPad Software, San Diego, CA, USA) for Mac. *p*-value ≤ 0.001: ***; *p*-value ≤ 0.01: **; *p*-value ≤ 0.05: *.

## 5. Conclusions

Targeted therapies have predominantly been designed to interfere with specific molecules that impact cancer cell proliferation and survival. However, often the targets expressed on the cancer cells are also found on neighboring stromal cells in the tumor microenvironment. Here, we provide the first evidence that the EGFR-targeted therapy cetuximab alters CAFs in a manner that protects CRC cells from the drug’s effects. These findings emphasize the importance of considering how targeted therapies may influence the microenvironmental milieu and ultimately alter tumor response.

## Figures and Tables

**Figure 1 cancers-12-01393-f001:**
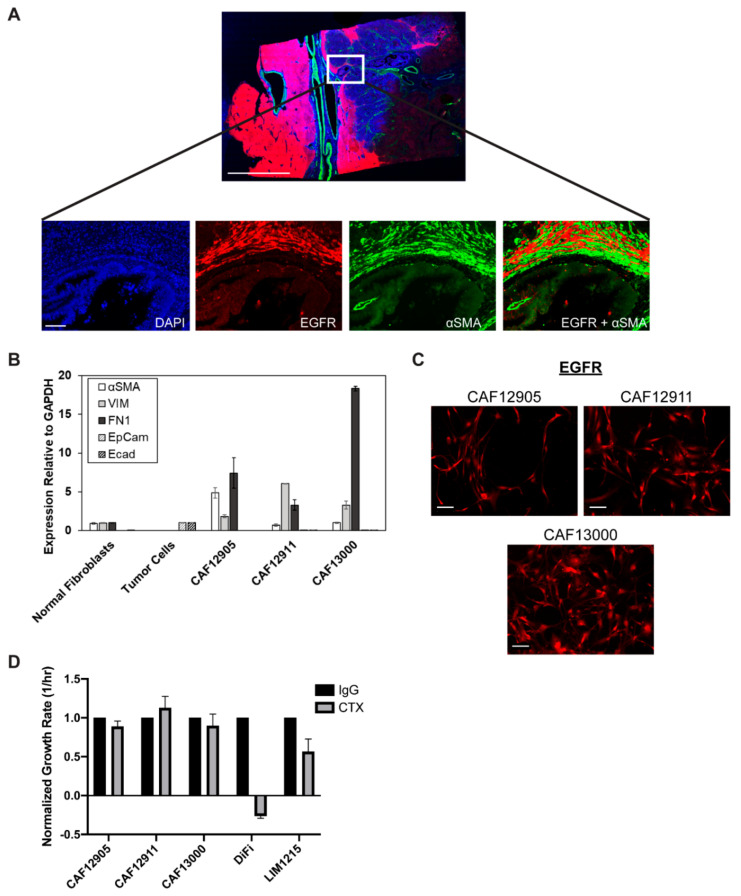
Cancer-associated fibroblasts (CAFs) express epidermal growth factor receptor (EGFR), but are not sensitive to EGFR inhibition. (**A**) Immunofluorescence stained colorectal cancer tissue from a biopsy of patient 12620 displayed expression of EGFR (red) and αSMA (green). Full tissue slice is shown on top (scale bar: 5 mm), with a 20× zoomed-in section shown below (scale bar: 100 μm). (**B**) Quantitative polymerase chain reaction (qPCR) analysis of CAF-associated markers alpha-smooth muscle actin (αSMA), fibronectin (FN1), and vimentin (VIM) was performed on primary cultured CAFs (isolated from colorectal cancer (CRC) tissues of patients 12,905, 12,911, 13,000), normal primary fibroblasts (NCF12737) and CRC tumor cells (LIM1215). (**C**) Expression of EGFR in CAFs 12,905, 12,911, and 13,000 as detected by immunofluorescence. Scale bar: 100 μm. (**D**) Cells were treated with 1 μg/mL cetuximab (CTX) or IgG control for five days. Live and dead cell counts were obtained on days 0, 3, and 5 to calculate growth rates (cell doubling per hour), which were normalized to IgG condition for each cell type.

**Figure 2 cancers-12-01393-f002:**
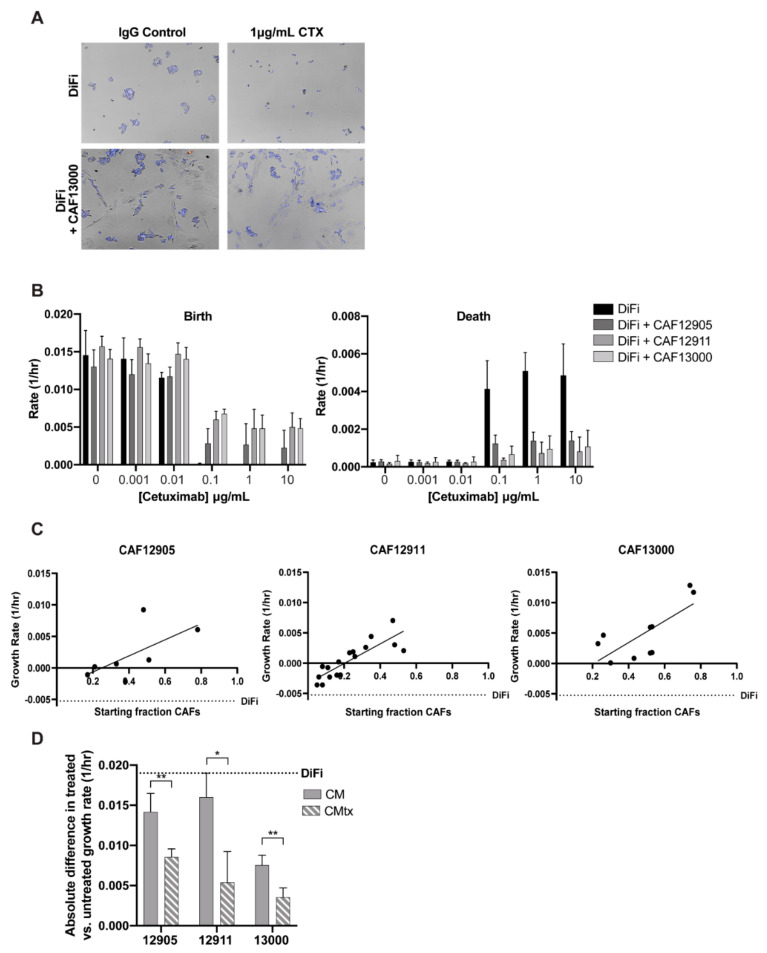
CAFs protect cancer cells from cetuximab treatment. CAFs and DiFi cancer cells were co-cultured and treated with various concentrations of cetuximab. (**A**) Representative images of DiFi and DiFi + CAF13000 co-culture treated with cetuximab or IgG control were taken five days post-treatment. (**B**) Birth (left) and death (right) rates of DiFi cells were calculated on co-cultures with CAF starting percentages ~50% by fitting live and dead cell counts taken on days 0, 3, and 5 to an exponential growth model. (**C**) Starting ratios of CAF and DiFi cells were calculated before a 5-day treatment with 1 μg/mL cetuximab and DiFi cell growth rates were calculated. The dotted line represents the growth rate of DiFi monoculture treated with 1 μg/mL cetuximab. Linear fits show an increasing slope, indicating increased tumor cell growth with increased CAF percentages upon cetuximab treatment. R^2^ values of fit: CAF12905 = 0.414; CAF12911 = 0.716; CAF13000 = 0.543. (**D**) Conditioned media was collected from CAFs untreated (CM) and treated with 1 μg/mL cetuximab (CMtx) after three days. DiFi cells were then cultured with the conditioned media conditions with or without cetuximab treatment for five days. The absolute difference between treated and untreated DiFi cell growth rates was calculated for each condition. The dotted line represents the absolute difference of DiFi monoculture. *p*-value ≤ 0.01: **; *p*-value ≤ 0.05: *.

**Figure 3 cancers-12-01393-f003:**
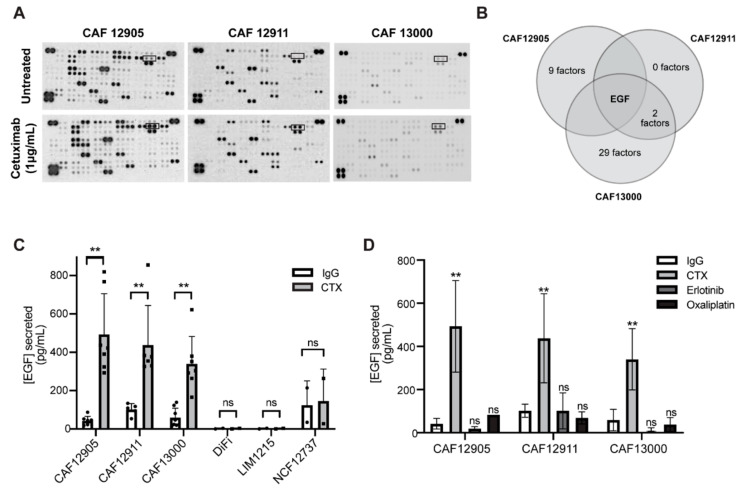
Cetuximab treatment alters CAF secretion profiles. (**A**) Raw images of cytokine array blots performed on conditioned media collected from CAFs treated with IgG control or 1 μg/mL cetuximab for 72 h. Boxed readings indicate epidermal growth factor (EGF). (**B**) Cytokine and growth factor expression was evaluated via cytokine arrays. The overlap of upregulated cytokines (>0.5 fold compared to untreated) across CAF lines is shown. (**C**) Levels of EGF secretion were determined via enzyme-linked immunosorbent assay (ELISA) on primary CAFs (12905, 12911, 13000), cancer cells (DiFi, LIM1215), and normal primary fibroblasts (NCF12737) lines. (**D**) Conditioned media was collected from CAFs treated with 1 μg/mL cetuximab, 1 μM erlotinib, 5 μM oxaliplatin, or 1 μg/mL IgG control for 72 h. ELISAs were performed to evaluate levels of EGF. *p*-value ≤ 0.01: **; *p*-value ≤ 0.05: *; ns: not significant.

**Figure 4 cancers-12-01393-f004:**
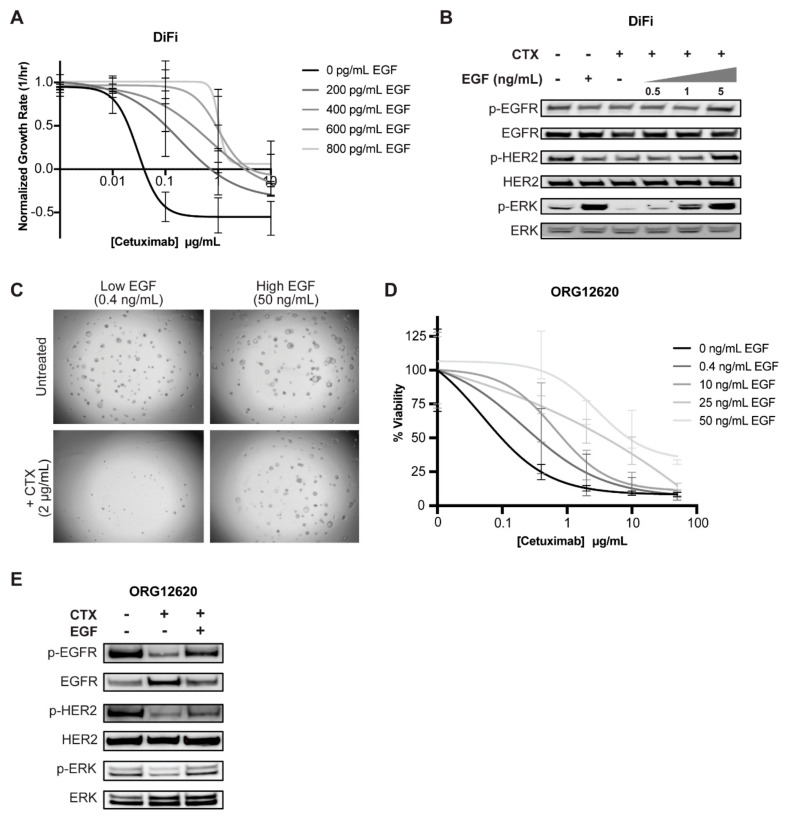
Exogenous EGF confers cetuximab resistance in cancer cell lines and organoids. (**A**) DiFi cells were treated with cetuximab in media containing various spike-in levels of EGF. Images were acquired on days 0, 3, and 5. Live and dead cell counts were obtained and fitted to an exponential growth model to calculate the growth rate. (**B**) After being serum-starved overnight, DiFi cells were treated with 10 µg/mL cetuximab and/or increasing concentrations of EGF for 2 h. Protein expression was evaluated by western blot. (**C,D**) Patient-derived colon tumor organoid line ORG12620 was treated with increasing concentrations of cetuximab in low EGF (0.4 ng/mL) patient-derived organoid (PDO) defined media for five days. (**C**) Images were acquired and (**D**) CellTiter-Glo was performed on organoids to determine percent viability. (**E**) ORG12620 were serum-starved overnight and treated with cetuximab (10 µg/mL) and/or EGF (100 ng/mL) for 2 h. Protein expression was evaluated via western blot.

**Figure 5 cancers-12-01393-f005:**
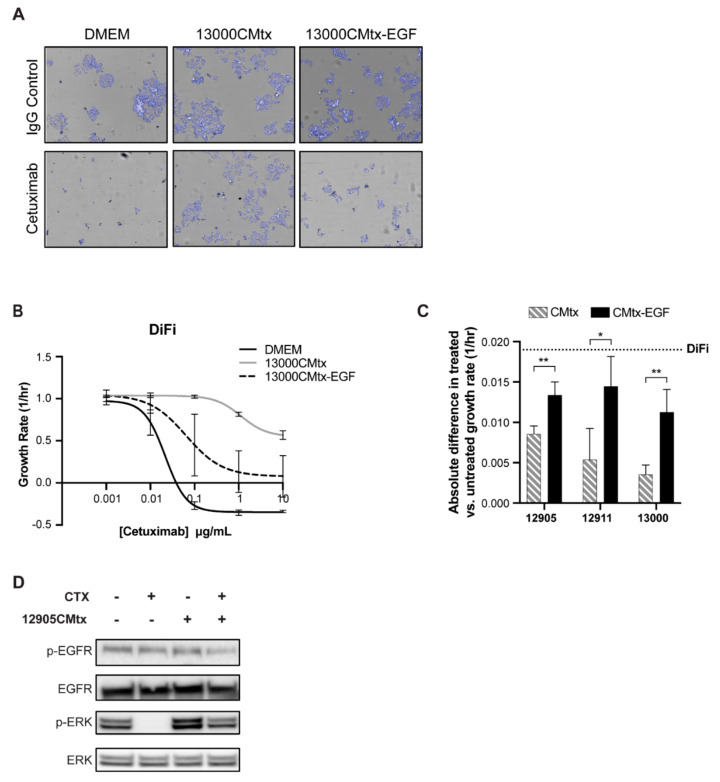
EGF is the factor in CAF CMtx conferring cetuximab resistance in cancer cells. (**A**) DiFi cells were treated with various cetuximab concentrations while cultured in Dulbecco’s Modified Eagle Media (DMEM), 13000CMtx, or 13000CMtx-EGF (i.e., 13000CMtx treated with anti-EGF) media. Images were acquired on days 0, 3, and 5, and representative images from day five are shown. (**B**) Live and dead cell counts were obtained and fitted to an exponential growth model to calculate the growth rate. (**C**) DiFi cells were cultured with CMtx or CMtx-EGF collected from CAF12905, CAF12911, and CAF13000 with or without 1 µg/mL cetuximab. Growth rates were calculated and the absolute difference (treated-untreated) is shown. (**D**) Conditioned media was collected from CAF12905 treated with 1 µg/mL cetuximab (12905CMtx) in fetal bovine serum (FBS)-free media. Following overnight serum-starving, DiFi cells were cultured with 1 µg/mL cetuximab and/or 12905CMtx for 2 h. Protein expression was evaluated via western blot. *p*-value ≤ 0.01: **; *p*-value ≤ 0.05: *.

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
