# Peer review of "Anti-EGFR Therapy Induces EGF Secretion by Cancer-Associated Fibroblasts to Confer Colorectal Cancer Chemoresistance"

_cancers, 2020, doi:10.3390/cancers12061393_

Round 1

Reviewer 1 Report

This manuscript has very interesting topic about tumor microenvironment factor induced anti-EGFR therapy resistance. Authors' hypothesis is cancer-associated fibroblast released EGF and the growth factor could block cetuximab effect and proved by some in vitro studies. this is very interesting manuscript, but still need more evidences to support authors' hypothesis.

Q1.  Some western blot data have to improve in this manuscript. Figure 4B shows not dramatic phospho-EGFR change depend on Cetuximab treatment.And, I could not understand Figure 4E and Figure 5D western blot data.  phospho EGFR may be inhibited by cetuximab and CMtx could not block cetuximab. 

Q2. Authors have to why used DiFi and LIM1215 cancer cell lines in this manuscript. Several previous reports recommended These cell lines shows high expression of EGFR and KRAS and BRAF wild type. for CAF release EGF related Cetuximab resistance, co-culture studies with these cells were good choice. Could Authors add more co-culture data using other Colorectal cancer cell lines? and KRAS mutation related effect also very important point. Until now, KRAS mutation status is major biomarker of EGFR targeted therapy. EGFR amplification and KRAS wild type colorectal cancer patients shows great response by Cetuximab therapy. So, additional study need to understand cetuximab resistance mechanism according to cancer associated fibroblast such as EGFR amp. vs EGFR low or KRAS wild vs KRAS mut. cancer cells. 

Q3. Co-culture study is great system to test the function of cancer associated fibroblast in vitro condition. However, in vivo study need for this manuscript. in vivo co--injection model also very useful and in vivo efficacy study results will be very supportive with current results for authors' hypothesis. 

Q4. Cancer associated fibroblast cell is very important. according to current results, authors have to show how to prepare the Cancer associated fibroblast cells with several data using molecular markers and descript more detailed information about these fibroblast cells. And, Authors have to think about consistent condition using fibroblast cell line (for example, TGF-beta induced CAF by NIH-3T3).  

Reviewer 2 Report

The manuscript entitled "Anti-EGFR Therapy Induces EGF Secretion by Cancer-Associated Fibroblasts to Confer Colorectal Cancer Chemoresistance" by Garvey et al inherited about the evaluation of cancer-associated fibroblasts role in the resistance to cetuximab treatment is well written and suitable for pubblication after moderate revisions

  • Please, could the authors better define the number of colorectal cancer patients involved in the study? In relation to this point, please, could they define if starting samples belong to the clinical patients? In my opinion, a moree exaustive description of preclinical models and clinical model samples should be implemented in the text.
  • In the supplementary tables, the authors report the molecular profile for the sampels adopted in  the study. The authors show that CAFs expressed several molecular alterations, and among of them, KRAS mutation was deteted. In relation to this evidence, could the authors express if this molecular alterations may influence EGF secretion after cetuximab treatment. Could the authors express if CAM models that do not harbour any alterations should be evaluated?
  • In the results section, the authors hypothesize that MAPK pathway represents the pivotal resistance mechanism to cetuximab induced by EGF secrection. Please, could the authors express if the adoption of a MAPK inhibitor could play a key role in the treatment of cetuximab resistance -fibroblasts associated?
  • In this manuscript the authors adopt cetuximab to text their hypothesis. Please, could the authors define of other mAB largel adopted in the clinical practice for wt mCRC patients (e.g. panitumumba) could have the same role in this approach?

Round 2

Reviewer 1 Report

Authors' kindly response and supplyment results are very interesting. And also helpful to understand CAF biology. I am fully agree about Human derived CAF resources are very tricky to maintain to use study. Human resources are very valuable for cancer study. I hope that more biological characterization about human CAF cells before testing. It will be very helpful to improve value of these cells. This manuscript still need to more in vivo study using cancer cell and CAF co-injection animal models. I hope that authors will be test using the co-injection animal models at further study.

Reviewer 2 Report

The manuscript entitled "Anti-EGFR therapy induces EGF secretion by cancer associated fibroblasts to confer colorectal cancer chemoresistance" is now available for pubblication without any additional revision.